# Fosfomycin Resistance in Bacteria Isolated from Companion Animals (Dogs and Cats)

**DOI:** 10.3390/vetsci10050337

**Published:** 2023-05-09

**Authors:** Marios Lysitsas, Irene Chatzipanagiotidou, Charalambos Billinis, George Valiakos

**Affiliations:** 1Faculty of Veterinary Science, University of Thessaly, 43100 Karditsa, Greece; lysitsas91@gmail.com (M.L.); billinis@uth.gr (C.B.); 2Department of Biochemistry and Biotechnology, University of Thessaly, 41500 Larissa, Greece

**Keywords:** Fosfomycin resistance, dog, cat, companion animals, pets, PRISMA guidelines, ARGs

## Abstract

**Simple Summary:**

Fosfomycin is an antibiotic with renewed interest over the last years, especially in human medicine, as it possesses some advantageous properties and a broad spectrum of bactericidal activity. Moreover, the emerging issue of multi-resistance against traditional antibiotics obligates specialists to turn to new alternative agents. However, cases of Fosfomycin-resistant strains are being detected at a rising rate worldwide, among humans and animals. The objective of this review is to collect, present and analyze studies related to Fosfomycin resistance in isolates from companion animals and specifically dogs and cats. Variable articles were collected and data for the relevant strains were scanned and evaluated. Since Fosfomycin is an agent not routinely used by veterinarians, the detection of Fosfomycin-resistant strains in canine and feline samples indicates possible dissemination of these strains among humans, pets, and the environment, reinforced by other factors. Concerning the origin, the species, and the resistance patterns of the related bacteria, useful conclusions were drawn, about the presence, the spreading, and possibly the causes of Fosfomycin resistance among companion animals and between them and their environment.

**Abstract:**

Fosfomycin is an old antibacterial agent, which is currently used mainly in human medicine, in uncomplicated Urinary Tract Infections (UTIs). The purpose of this review is to investigate the presence and the characteristics of Fosfomycin resistance in bacteria isolated from canine or feline samples, estimate the possible causes of the dissemination of associated strains in pets, and underline the requirements of prospective relevant studies. Preferred Reporting Items for Systematic Reviews (PRISMA) guidelines were used for the search of current literature in two databases. A total of 33 articles were finally included in the review. Relevant data were tracked down, assembled, and compared. Referring to the geographical distribution, Northeast Asia was the main area of origin of the studies. *E. coli* was the predominant species detected, followed by other Enterobacteriaceae, Staphylococci, and *Pseudomonas* spp. *FosA* and *fosA3* were the more frequently encountered Antimicrobial Resistance Genes (ARGs) in the related Gram-negative isolates, while *fosB* was regularly encountered in Gram-positive ones. The majority of the strains were multidrug-resistant (MDR) and co-carried resistance genes against several classes of antibiotics and especially β-Lactams, such as *bla_CTX-M_* and *mecA*. These results demonstrate the fact that the cause of the spreading of Fosfomycin-resistant bacteria among pets could be the extended use of other antibacterial agents, that promote the prevalence of MDR, epidemic strains among an animal population. Through the circulation of these strains into a community, a public health issue could arise. Further research is essential though, for the comprehensive consideration of the issue, as the current data are limited.

## 1. Introduction

### 1.1. Properties and Mode of Action of Fosfomycin

Fosfomycin is a hydrophilic, low molecular mass (138 Da) derivative of a phosphoric acid, which was first discovered in Spain in 1969, under the name phosphonomycin [1]. It has been isolated from cultures of *Streptomyces* spp. [2,3]. It remains the only antibiotic in its class, and thus its role as an alternative therapeutic option is unquestionably important [4].

Fosfomycin’s bactericidal effect is achieved by blocking the first step of peptidoglycan synthesis, which is the structural unit of the bacterial cell wall, resulting in the inhibition of its biogenesis and consequently lysis of the bacterial cell [5]. The transport of Fosfomycin to the interior of the bacteria is performed through specific permeases (membrane transport proteins that subserve the diffusion of specific molecules intracellularly), which are the glycerol-3-phosphate transporter (GlpT) and the glucose-6-phosphate transporter (UhpT) [4]. Once diffused intracellularly, it inhibits the UDP-N-acetylglucosamine enolpyruvyl transferase (MurA) enzyme, which is responsible for catalyzing the formation of N-acetylmuramic acid, peptidoglycan’s precursor. Therefore, it inhibits cell wall formation by blocking the first step of its biosynthesis [5].

Fosfomycin is an antibacterial agent of broad spectrum and very low toxicity. Additionally, it has a very low protein binding (0,5%) and is highly distributed throughout the body, including inflamed tissues and pus [6]. Several studies confirm that Fosfomycin has the ability to penetrate into tissues where antibiotics frequently demonstrate low penetration ability; therefore, it has a significant potential for usage against many difficult-to-treat infections [3,7]. Another major advantage is its diminished cross-resistance property, as its mechanism of action is unique, resulting in an absence of crossed resistances with other classes of antibiotics [4].

### 1.2. Usage in Human and Veterinary Medicine

Oral Fosfomycin is primarily used in human medicine, in cases of uncomplicated urinary tract infections (UTIs) and prostatitis caused by multidrug-resistant (MDR) Gram-negative bacteria. Intravenous Fosfomycin has been mainly used in combination with other agents against various types of complicated, severe MDR infections [8].

In recent years, however, the use of Fosfomycin has increased spectacularly due to the considerable incidence of MDR microorganisms, against which it constitutes (alone or in combination) a treatment alternative [3,9]. It has a broad spectrum of in vitro activity against a variety of pathogens, such as methicillin-resistant *Staphylococcus aureus* (MRSA), drug-resistant Enterobacteriaceae, and *Pseudomonas aeruginosa* including extended-spectrum-β-lactamase (ESBL)-producing and carbapenem-resistant (CR) organisms [10,11,12]. Different studies also confirm the in vivo and in vitro synergistic action of Fosfomycin in combination with other antibiotics, against MDR *Acinetobacter baumannii* [13,14,15] and Vancomycin-resistant Enterococci (VRE) strains [16,17].

In veterinary medicine, Fosfomycin is used mostly against infections caused by a number of Gram-positive and Gram-negative pathogens, including *E. coli*, in intensively bred piglets and broiler chickens [3,18]. Especially in piglets, Fosfomycin is effective in the treatment of several stress-associated or secondary (due to the dissemination of viruses) bacterial infections [18]. It is more widely used in farms in South and Central America. Even though it is occasionally used by veterinarians for decades and in several countries, it Is still considered a second-line antibiotic. It is not approved for veterinary use in many countries, and therefore knowledge of its pharmacokinetic and pharmacodynamic properties among professionals is rather insufficient [19]. Moreover, it is currently categorized as a critically important antimicrobial for human medicine and thus its use is rather disapproved, especially in European Union [20,21]. Nevertheless, a small number of protocols have been suggested, with promiscuous results, in several livestock and domestic species such as broiler chickens [22,23], dogs [19,24], pigs [25,26], cattle [27], and horses [28]. Contrarily, acute renal insufficiency and failure have been referred to after administration in cats [29,30].

### 1.3. Fosfomycin Resistance

Even though Fosfomycin seems to be still effective against a great variety of bacteria, several resistance mechanisms have been described. The main mechanisms are described in Table 1.

### 1.4. Objectives of the Review

The objective of this review is to collect, present and analyze the results of studies related to Fosfomycin resistance in bacteria isolated from dogs and cats, on an international level. Moreover, the objective is to evaluate the available data for the associated bacterial species, their phenotypic and molecular resistance profiles, and their dissemination in animal populations, in order to determine possible causes for the prevalence of Fosfomycin resistance in pets.

The final objective is to underline possible emerging concerns regarding public health and, considering these concerns, recommend potential requirements for future research in this area of interest, as the current data are rather insufficient for a circumstantial comprehension of the issue.

## 2. Materials and Methods

The Preferred Reporting Items for Systematic Reviews (PRISMA) guidelines were evaluated for this study [47]. The individual steps of the process are presented in Figure 1.

Initially, we searched for reviews on the subject of Fosfomycin resistance in companion animals at an international level in the following databases: Google Scholar and PubMed. In these databases, 1821 studies were found using the keywords Fosfomycin and resistance and extra keywords, such as companion animals, dog, cat, canine, feline, and pet, alone or in variable combinations.

The selected studies were published in peer-reviewed journals, websites of organizations, books, and dissertations mostly in the English language. The first step was a screening based on the titles of the articles. We excluded all those not related such as duplicates and studies referring to human medicine, pharmacodynamics, pharmacokinetics and other antibacterial agents, microbiology and genetics, wild animals, water, and environment, and studies referring to food, livestock, and other domestic animals except dogs and cats. During the second selection phase, the abstracts of the reviewed studies were examined fully and independently, in order to identify their relevance according to the information that was searched. Generic information was collected from each article, such as the author, year of publication, the country where the study was conducted, its design and unit of interest, and number of subjects.

Specifically, a total of 1821 manuscripts were finally found: 391 in PubMed and 1430 in Google Scholar. A total of 1035 publications were first excluded as their title was fully irrelevant or they were duplicates or they did not provide an abstract or they were written exclusively in a different language. Subsequently, of the remaining 796 articles whose abstracts were examined, 529 were rejected because their abstracts were irrelevant to the scope of this review, according to the criteria referred to previously. For these reasons, 267 manuscripts remained and 16 could not be retrieved. Therefore, 251 studies were left to be examined and, among them, we rejected 87 as they only concerned human medicine, 30 as they only concerned microbiology and genetics; 31 as they only concerned pharmacodynamics, pharmacokinetics, or different antibacterial agents; 18 as they only concerned wild animals, water or the environment; and 52 as they only concerned food, livestock, or other domestic animals. Finally, 33 manuscripts were used in this review.

The public information extracted from each of the selected articles is presented in Table 2. The country/area of isolation, the bacterial species, the ARGs (if referred), the origin of the sample (dogs, cats, both), the number of fosfomycin-resistant isolates reported (with associated genes recognized in parenthesis), and the date/time period of isolation are listed. Statistical analysis included descriptive analysis of the prevalence of fosfomycin-resistant bacterial strains reported in the various manuscripts significant, comparing the findings among bacterial species, genes detected, and countries reporting results.

## 3. Results

### 3.1. Geographical Distribution

This distribution of the isolates included in this review is visualized in Figure 2. Regarding the continental distribution, Asia, North, South America, and Europe hold the grand majority, whereas Africa and Oceania are hardly represented.

### 3.2. Relevant Findings of the Studies Per Country of Isolates Origin

#### 3.2.1. China

In a study implemented in Guangdong province [50], 323 *E. coli* strains were tested for Fosfomycin resistance. They were isolated from living and diseased pets (dogs and cats), from 2006 to 2010. A total of 33 isolates (10.2%) were resistant, while 29 of them carried the ARG *fosA3*. The presence of resistant bacteria while the antibacterial agent was not administered to any of these pets, could be a result of the co-selection of the *fosA3* gene with other ARGs, after the extended use of aminoglycosides and/or cephalosporines and dissemination through the horizontal transfer of plasmids [50].

One of the strains of the previous study, recovered from a dog in Guangzhou, in 2008, was further investigated. Plasmid DNA was fully sequenced and it was proved that the *fosA3* gene was located in a Multiresistance Region (MRR) of a plasmid, together with aminoglycoside-resistance gene *rmtB* and ESBL gene *bla_CTX-M-65_* [51]. This enhances the hypothesis that the dissemination of resistance to Fosfomycin could be achieved through epidemic plasmids, under the pressure of other antibacterial agents, which are routinely used in companion animals.

During the period of September 2008–December 2010, a total of 2106 fecal animal samples were collected from the area of Hong Kong and, among them, 368 were from stray dogs and 398 from stray cats. Nine Fosfomycin-resistant strains were found in dogs and three in cats and the *fosA3* gene was detected in all but one canine isolate. Additionally, an interesting fact was that Fosfomycin-resistant isolates were more likely to be MDR than susceptible ones. Finally, the results, in concordance with the previous studies, demonstrated the dissemination of *fosA3*-mediated resistance among MDR bacteria, in several domestic and wild animals [52].

In another study, 171 samples of pets and their owners were collected in a veterinary Teaching Hospital in Beijing, China, from March to June 2013. All of the samples were inoculated in media supplemented with Fosfomycin in order to isolate resistant strains. Nineteen resistant Enterobacteriaceae isolates were detected. Among them, 16 strains were *fosA3* positive (8 *E. coli*, 4 *P. mirabilis*, 3 *E. fergusonii*, and 1 *C. freundii)* and three were *fosA* positive (*Enterobacter aerogenes*, *Klebsiella oxytoca*, *Klebsiella pneumoniae*). All of them were MDR. Two different genetic environments of the *fosA3* gene were detected, both related to plasmids previously studied, from bacteria of human and swine origin. Therefore, the danger of transmission of *fosA3*-carrying plasmids or clones between humans and animals is indicated [56].

An *E. coli* strain isolated from a dog, in an Animal Teaching Hospital (Beijing), in 2013, during a surveillance study with samples originating from companion animals, was found positive for the *bla_NDM-1_* gene, that confers resistance to carbapenems. This isolate was also highly resistant to Fosfomycin among other antibacterial agents [63].

From July 2017 to October 2019, 5359 samples of companion animals were collected in Beijing and Tianjin, from which 105 *Klebsiella pneumoniae* strains were isolated. Even though Fosfomycin was not among the antibiotics tested in the AST, the *fosA* gene was detected in all the isolates [72].

In Shanghai, 79 samples were collected from 31 hospitalized pets in a veterinary hospital, from May to July 2021. Seven *E. coli* isolates positive for the colistin-resistance gene *mcr-1* were detected. All of them carried the *fosA3* gene and also exhibited an MDR phenotype [76]. The spread of the *mcr-1* gene in companion animals was associated with plasmid transmission, horizontal and vertical. The *fosA3* gene was also related, as it was co-carried in the same plasmid with the *mcr-1* in all isolates, and both Colistin and Fosfomycin are agents of limited use in dogs and cats.

Finally, in a study conducted in Zhejiang, canine and feline fecal samples were collected and *Salmonella enterica* isolates were obtained. A high percentage of these bacteria was MDR and several ARGs were identified. Among them, Fosfomycin-resistance gene *fosX* was detected in a number of *Salmonella* Dublin strains [80].

#### 3.2.2. USA

In a study, 200 clinical and 75 experimental strains of *E. coli* were isolated from samples of canine and feline origin, from March 2008 to December 2010. The minimum inhibitory concentration of Fosfomycin and six other classes of antibacterial agents was determined. Only 3/275 isolates were non-susceptible to Fosfomycin, while according to the antimicrobial susceptibility testing (AST) phenotype, they were categorized as: resistant to no drugs (NDR): 47 strains, a single drug class resistance (SDR): 88 strains, 3–4 drug classes resistance (MDR): 112 strains and 5–6 drug classes resistant (Extensively Drug Resistant, XDR): 18 strains. These data suggested that, even though there was a very limited rate of non-susceptibility, Fosfomycin was undoubtedly very effective against *E. coli* related to disease in dogs and cats. Moreover, a large number of MDR and XDR strains were identified. These in vitro results though, require further research for dosing, clinical safety, and efficacy to be determined [48].

Fecal samples of 554 dogs from seven animal shelters across Texas were collected from May 2013 to December 2014 and tested for the detection of *Salmonella* spp. The 27 *Salmonella enterica* isolates were sequenced and screened for ARGs. Two of them (serotypes Heidelberg and Derby) carried the Fosfomycin-resistance gene *fosA7* [71].

In a study from China, a comprehensive comparative analysis of 66 genomes of *Micrococcus luteus*, downloaded from the National Center for Biotechnology Information (NCBI) GenBank database was performed. Among them, a strain isolated from a sample of canine ocular discharge, originating from the USA (New Hampshire) carried the *murA* gene, associated with Fosfomycin resistance. This gene was identified in all 66 genomes [75].

Finally, *Staphylococcus aureus* isolates from clinical animal specimens were collected in New England, from September 2017 through March 2020. Using whole genome sequencing, the distribution of acquired genes related to antibiotic resistance was searched in 53 genomes of canine or feline origin, among other tasks. The Fosfomycin-resistance gene *fosB* was identified in 42 genomes of canine or feline origin [79].

#### 3.2.3. Canada

A total of 542 Enterobacteriaceae strains were isolated from canine clinical samples from two diagnostic laboratories in Ontario, from November 2015 to October 2016. They were subjected to AST and PCR screening for ARGs. The genome of all *bla _CTX-M_* positive bacteria was sequenced (a total of 47 isolates). Fosfomycin-resistance gene *fosA* was detected in two *Klebsiella pneumoniae* and one *Enterobacter cloacae* isolate [59].

R. M. Courtice [65] searched the prevalence of Fosfomycin resistance in 387 canine urine isolates, including 274 *E. coli* and 113 *S. pseudintermedius*. Of these strains, 11 were resistant (seven *E. coli* and four *S. pseudintermedius* strains). Among other results, a statistically significant relationship between methicillin and Fosfomycin resistance in *Staphylococcus pseudintermedius* was observed, as three of four Methicillin-Resistant *Staphylococcus pseudintermedius* (MRSP) detected among the 113 isolates were also Fosfomycin-resistant. It is noted that this should be estimated with caution, due to the small number of MRSP isolates. Additionally, as ARGs *fosA*, *fosA3*, *fosB*, and *fosC* were not identified in any of the resistant strains, it was suggested that an alternative mechanism such as mutations in *murA*, *glpT*, *uhpA*, or *uhpT* genes could be the cause of the resistance.

#### 3.2.4. USA and Canada

Isolates of MRSP, from Canada (21 strains) and the USA (10 strains), were screened for Fosfomycin resistance and for the presence of the ARG *fosB*. Only seven strains were resistant while 27 of them (all of the resistant and 20 of the susceptible) carried the gene. It was estimated that this could be a result of repression or a non-functional *fosB* and the need for further study of Fosfomycin resistance in MRSPs was noted [53].

#### 3.2.5. Germany

In a study in Germany, 223 clinical isolates of *Acinetobacter baumannii* were obtained from veterinary clinics between 2000 and 2013. They were screened for carbapenem non-susceptibility and AST for several antibiotics was also performed. A subgroup of 25 strains was afterward tested against some extra antibacterial agents including Fosfomycin. All of them (25/25) were Fosfomycin-resistant [55].

Between May 2015 and March 2016, bacteria were obtained from samples of companion animals. Antimicrobial susceptibility testing was performed. Staphylococci were screened for Methicillin Resistance and Enterobacteriaceae for ESBL production, by phenotypic methods and PCR. Four Staphylococci isolates were resistant to Fosfomycin; three were derived from rabbits and a *Staphylococcus cohnii* subsp. *cohnii* isolate originated from a canine buccal sample [60].

In another study that took place in China, a *Salmonella enterica* serovar Telelkebir clinical isolate and 120 available genomes obtained from databases, were investigated for variable factors, such as their relatedness, ARGs, and virulence factors. Among them, there was an isolate from a companion animal in Germany (2007), which carried Fosfomycin-resistance gene *fosA7* [73].

#### 3.2.6. Brazil

In December 2016, a *Pseudomonas aeruginosa* isolate was recovered from the ear canal of a dog in a veterinary clinic in Brazil. This strain harbored many ARGs for several classes of antibacterial agents including β-Lactams and Carbapenems, Aminoglycosides, Tetracyclines, Folate Pathway Inhibitors, Phenicols, and Fosfomycin. More specifically, concerning Fosfomycin resistance, the *fosA* gene was detected. Furthermore, as the same strain was isolated from the pet owner, who has a recent event of hospitalization and the house environment, it was proved that household dissemination occurred. This fact indicated the danger of transmission of hospital-acquired MDR pathogens from humans to companion animals and the circulation of them in a household environment [62].

Another case of an MDR isolate, carrying a Fosfomycin-resistance gene (*fosA*), occurred in 2018. In a veterinary hospital, a *Klebsiella pneumoniae* strain was obtained from a urinary tract infection (UTI) of a 6-year-old dog. Antimicrobial susceptibility testing and genomic DNA sequencing were performed. Antimicrobial resistance genes were identified against several classes of antibiotics and *fosA* was one of them, although by disc diffusion method, a susceptible phenotype for Fosfomycin had been demonstrated [67].

In March 2019, in a veterinary teaching hospital in southeast Brazil, a *Klebsiella pneumoniae* strain was isolated from a dog with otitis. Antimicrobial susceptibility testing and whole genome sequencing were performed. *FosA* was detected among variable ARGs, after resistome analysis, even though the strain was phenotypically susceptible to the agent [68].

#### 3.2.7. France

In a study conducted in France, coagulase-positive Staphylococci isolated from canine and feline samples, from 2006 to 2010, were tested for Methicillin resistance. A group of 23 *MRSA* was obtained. Subsequently, susceptibility testing and molecular typing were carried out. In 19 of 23 strains, Fosfomycin-resistance gene *fosB* was identified. Most of these strains were MDR and related to human MRSA clones [49].

From 2008 to 2011, 68 *Pseudomonas aeruginosa* clinical isolates of variable animals (dogs, dairy cows, horses) were collected. Genotyping and AST were performed. A rate of 47.8% (22/46) of Fosfomycin resistance was observed in canine isolates. Additionally, among these isolates, a common combination of resistances noticed was the β-Lactams-Aminoglycosides-Fosfomycin [54].

#### 3.2.8. Japan

From September 2014 to February 2015, 200 *Pseudomonas aeruginosa* strains were obtained from samples of dogs and cats, in veterinary hospitals located in several areas of Japan. Antimicrobial susceptibility testing and screening for aminoglycoside-resistance enzymes and metallo-β-lactamases were performed. Resistance against Fosfomycin was observed in 35.5% of the isolates, the highest among the tested agents, as rates of resistance were generally low [57].

In another study, isolates from specimens of dogs and cats were collected between May and September 2016, in clinical settings and hospitals throughout Japan. A total of 381 *E. coli*, 50 *Klebsiella pneumoniae*, and 56 *Proteus mirabilis* were obtained. AST and a multilevel molecular analysis were performed. Plasmid-mediated Fosfomycin-resistance gene *fosA3* was detected in two *E. coli* strains, co-harboring *bla_CTX-M-55_* plus *bla_CMY-2_* and *bla_CTX-M-27_* genes, respectively, and in a *Klebsiella pneumoniae* strain co-harboring *bla_SHV-12_*, *bla_DHA-1_*, and *armA* [58].

#### 3.2.9. Ecuador

In Quito, 50 canine fecal samples were collected in August 2017 and inoculated in ceftriaxone-supplemented agar, to enhance the growth of strains resistant to the agent. A total of 23 *E. coli* isolates were obtained. AST and genotyping characterization of the bacteria were subsequently carried out. Resistance for Fosfomycin was observed at 43% of the strains (10/23). Additionally, all of them were MDR, and a very high resistance rate was detected for most of the antibiotics tested [64].

For another study, 42 samples were collected from backyard animals in June 2016. They were plated to selective media in order to isolate Colistin-resistant strains. Three isolates were obtained, two of canine origin. Among other tasks, the Fosfomycin-resistance gene *fosA3* was detected [70].

#### 3.2.10. India

An XDR-resistant *E. coli* was isolated from the scrotal fluid of a 3-year-old Labrador, in a veterinary hospital. Antimicrobial susceptibility testing and phenotypic and molecular screening for ARGs were accomplished and the strain was resistant against almost every tested agent (except Tigecycline), including Fosfomycin. Variable ARGs for β-Lactams, including a *bla_NDM1_* carbapenemase-producing gene, were detected [61].

#### 3.2.11. Australia

A study was carried out in Melbourne, between November 2016 and May 2017. Samples of environmental sources of an animal Hospital were collected. The microbial population was evaluated and screened for ARGs, before further molecular investigation. Antimicrobial resistance genes against Fosfomycin were detected in Enterobacteriaceae species from all four sources of the samples (Intensive Care Unit (ICU) cages, Laundry Trolley, Mop Bucket, and Office Corridor) and in *Pseudomonas* spp. from ICU cages. *FosA* detection is specifically referred to in *Klebsiella* and *Enterobacter* species, located on chromosomal sequences [66].

#### 3.2.12. Taiwan

A group of 19 *Enterobacter cloacae* isolates was obtained from UTIs of companion animals (10 dogs and 9 cats) in a Veterinary Teaching Hospital. Antimicrobial resistance genes for Fosfomycin and co-existing resistance genes were screened, and AST and conjugation experiments were also performed. The *fosA* gene was present in eight strains, three of which co-carried the *fosA3* gene. *FosA* was likely located on the chromosome, while *fosA3* was on mobile genetic elements [69].

#### 3.2.13. Hungary

During a study, a group of 102 dogs and 84 owners was sampled in Budapest and 14 other towns. Subsequently, 27 *S. aureus* and 58 *S. pseudintermedius* isolates were obtained. AST and molecular investigation of the strains occurred. In a pair of *S. aureus* isolates, from a dog and its owner, the *fosB* gene was present. In another pair of *S. aureus* isolates, from a dog and another owner, amino acid changes in *murA* transferase and *glpT* transporter were detected. These changes can confer Fosfomycin resistance by different mechanisms [74].

#### 3.2.14. Caribbean

For a study carried out on St. Kitts, a small Caribbean island, 82 *Klebsiella* spp. strains were collected from several sources (human, canine, feline, equine, vervet). A whole genome sequence analysis was accomplished. The *fosA* gene was detected in a *Klebsiella pneumoniae* of canine origin and one of feline origin. It is noted in the study that this gene belongs to the core genome of *Klebsiella* and therefore it should not be regarded as acquired [77].

#### 3.2.15. South Africa

In Pretoria, an *E. coli* strain was isolated from a deceased dog. Antimicrobial susceptibility testing was performed by both disc diffusion and MIC. The isolate was MDR and possessed virulence factors for two pathotypes. It was proved to be Fosfomycin-resistant in the MIC test, even though in the disk diffusion test it demonstrated a susceptible phenotype [78].

### 3.3. Microorganisms and ARGs of Interest

The isolated bacteria related to Fosfomycin Resistance in the selected articles and the associated ARGs are presented in Table 3.

*E. coli* is more frequently referred to (13/33, 39% of studies), followed by *Kl. pneumoniae* (7/33, 21% of studies), while the Enterobacteriaceae family is strongly represented by many more of its members (8/33, 24% of studies). These bacteria are mostly associated (when searched) with the presence of *fosA* (including *fosA^SH^*) and *fosA3* genes. Staphylococci and *Pseudomonas aeruginosa* also make their appearance more than once (6/33 18% and 4/33, 12% of studies, respectively), usually related to *fosB* and *fosA* genes, respectively. Finally, to our knowledge, there is only one relevant reference for *Micrococcus luteus* and for *Acinetobacter baumannii*.

In reference to the molecular basis of the resistance, the *fosA* gene is the most frequently detected ARG with nine references plus two references of *fosA^SH^* (*fosA7*). *FosA3* is present in eight of the studies, almost all of them (7/8) located in southeast Asia and Japan. *FosB* is identified in four articles, from Staphylococci (*S. aureus* and *S. pseudintermedius*) in Europe and North America (two articles, respectively). There are only two references to the detection of the *murA* gene and one of the *glpT* and *fosX*, in the selected literature.

### 3.4. Resistome and Phenotypic Resistance against Other Antibacterial Agents in Fosfomycin-Resistant Strains

The selected studies contain a large amount of information about susceptibility testing and different mechanisms of resistance of the associated bacteria, such as acquired, intrinsic, silent, and protoresistance (little/no activity until mutated). All these mechanisms are described by the term resistome, as it has been previously described [81]. In order to evaluate the resistome of the isolates related to Fosfomycin resistance, data of their ARGs and phenotypic susceptibility testing were collected and presented in Table 4. Some articles have been excluded due to difficulties in the classification of the available information.

The majority of the correlated strains are MDR bacteria, as indicated by the number of antibacterial agents with >50% resistance rate among them, presented in Table 4. Furthermore, there is a notable presence of methicillin resistance in the related Staphylococci (4/5 articles) and a notable presence of ESBL genes in the related Enterobacteriaceae, with *bla_CTX-M_* gene detected in at least 11 studies and more β-Lactamase encoding genes such as *bla_SHV_*, *bla_OXA_*, *bla_TEM_*, even *bla_NDM-1_* referred in variable occasions. Other classes of antibiotics are also included several times in the resistome of the isolates, such as aminoglycosides, quinolones, tetracyclines, folate pathway inhibitors, phenicols, polymyxins, etc.

## 4. Discussion

In this review, several cases of Fosfomycin resistance in bacteria from companion animals were presented, even though the available data are relatively limited and difficult to evaluate at present. However, the emergence of higher resistance rates would definitely constitute a challenge for veterinarians and a concern for public health, as this antibiotic demonstrates potential as an alternative agent in the upcoming era of bacterial multi-resistance [3,6,10].

Moreover, the dissemination of Fosfomycin resistance is already a matter of concern in human medicine. The *fosA3* gene is rather endemic in Southeast Asia. The detection of this resistance gene in plasmids of MDR clones, co-carrying other ARGs, indicates the danger of higher rates of resistance, under the pressure of antibacterial agents widely used, especially β-Lactams [5,82]. Importation of isolates carrying similar plasmids in Europe has also been reported [83], demonstrating the danger of universal spreading.

Regarding companion animals, even if the use of Fosfomycin is relatively limited in many countries, cases of phenotypic resistance and correlated ARGs are referred to in the studies included in this review. Pets, especially dogs and cats, could contribute to the prevalence of resistant isolates in a community, considering their household accommodation, their close contact with their owners, their large numbers in urban areas, and the possibility of hospital-acquired MDR infections in veterinary hospitals. The danger of a pet–owner circulation of resistant bacteria is a subject of research in some of the included articles [56,62,70,74] and a matter of concern in the grand majority of them. Additionally, hospital-acquired infections are present in small animal practices, among veterinary hospitals of variable sizes and it is estimated that their frequency is going to increase [84].

Moreover, the real occurrence of Fosfomycin resistance in pets is undoubtedly expected to be more significant, considering that several facts could influence its current sub-detection. Specifically, Fosfomycin is a rarely used agent in the AST of isolates from companion animals in many countries, especially in bacteria not originated from UTIs, and therefore a phenotypic resistance could be frequently missed. ARGs for Fosfomycin are also not usually screened in molecular assays and, consequently, they are not detected. A characteristic of the limited available data is that only one of the articles included in this review originated from Africa [78] and the references from Europe come only from three countries (France, Germany, Hungary) [49,54,55,60,73,74]. More relevant studies have been conducted in Southeast Asia (Figure 2). Furthermore, only a few of the 33 included studies were particularly aiming to search for phenotypic Fosfomycin resistance or correlated ARGs [48,50,51,52,53,56,65,69], whereas, in most cases, the detection was random during investigations.

Difficulties may occur in regard to the interpretation of Fosfomycin resistance. The MIC and the disk diffusion methods for Fosfomycin have their own specifications and should be evaluated with caution [85]. Furthermore, in the included studies, bacteria carrying ARGs for Fosfomycin demonstrated a susceptible AST in some cases [53,67,68], while there is an ambiguous AST, with an isolate observed to be susceptible by the disk diffusion method and proved to be resistant by the MIC method [78]. Additionally, the existing clinical breakpoints refer exclusively to human medicine with only a few exceptions [86], and the clinical trials in companion animals are limited [18]. Therefore, only an estimation can be formulated about the effectiveness of the drug in vivo, by the evaluation of the in vitro susceptibility testing.

As a result, the lack of sufficient relevant data and the demanding assessment of the existing information were limiting factors for this current review.

However, a shaky interest in Fosfomycin might emerge lately, even in veterinary medicine. This can be demonstrated by the fact that all of the relative articles included in this study have been completed over the last two decades and published after 2010. As a consequence, a more comprehensive evaluation of Fosfomycin resistance in companion animals is expected to be possible during the following years.

Nevertheless, some interesting facts can be estimated, even from the current data. Initially, Fosfomycin resistance in dogs and cats has mostly been searched in specific areas of the World, such as Southeast Asia, America, and part of Europe. The lack of relevant data from other areas is not necessarily a result of full susceptibility, as it is usually not included in the AST. Moreover, the interpretation of the resistance is a challenging task and in vitro AST tests or molecular assays might not always represent the clinical effectiveness of the drug.

Enterobacteriaceae are the isolates mainly associated with Fosfomycin resistance in dogs and cats (Table 3). This was anticipated, considering their wide dissemination, their significance as pathogens, and the increased interest in their resistance mechanisms in the current research fields. *FosA* and *fosA3* are the dominant relevant ARGs in Gram-negative bacteria, located in chromosomal DNA and plasmids, respectively. The location of *fosA3* gene in mobile genetic elements creates concerns about the wide dissemination of the resistance through the transmission of these elements among different bacterial strains [50,51,52,56,58,69,70,76]. Acquired *fosA3*-mediated resistance is the dominant mechanism detected in *E. coli* isolates, while it is only occasionally detected [56,58] in other bacterial species of the Enterobacteriaceae family. Furthermore, *fosA3* and *bla_CTX-M_* are regularly co-carried [50,51,52,56,70,76] and possibly co-transferred through common mobile genetic elements. *FosB* is the main ARG identified in Gram-positive cocci [49,53,74,79].

Concerning the prevalence of resistance among the total number of isolates included in the selected studies, *Acinetobacter baumannii* (25/25 isolates, 100%) and *Pseudomonas* spp. (117/306 isolates, 38%) are exhibiting the highest resistance rates, even though they are more infrequently detected than Enterobacteriaceae. This fact is unsurprising, as these species demonstrate reduced susceptibility through inherent mechanisms [3,10].

The connection between Fosfomycin resistance and MDR strains is beyond a doubt a fact, especially in Enterobacteriaceae and Staphylococci, as is clearly indicated by Table 4. Particularly, ARGs and phenotypic resistances for β-Lactams are detected in the grand majority of the studies. In combination with the infrequent use of Fosfomycin, this fact indicates the possible prevalence of Fosfomycin-resistant bacteria under the pressure of wide usage of antibacterial agents belonging to this class. Several other antibiotics make their appearance regularly in Table 4, suggesting that more routinely used agents could also have an impact, subserving the dominance of resistant strains. Aminoglycosides and tetracyclines are the most frequently observed classes. This co-selection issue is highlighted in a number of the included studies [50,51,52,76]. Moreover, recent studies have identified the colonization of pets by MDR bacteria after antibiotic treatment with commonly used drugs, such as β-Lactams and Fluoroquinolones, and thus the requirement for increased surveillance efforts [87,88].

Consideration of the aforementioned facts underlines the necessity of future research. The agenda of the prospective studies could include surveillance studies in countries/areas where there are no current data for Fosfomycin resistance in veterinary samples, in order to determine the presence of resistant bacteria and their phenotypic and genotypic characteristics. Additionally, a correlation of the presence of resistant strains, with previous antibiotic treatments of the animal, evaluating all the available data (such as the number of treatments during the preceding time period, classes of antibiotics received, duration of the therapy, etc.), would be of major significance in the assessment of antibiotic usage as a predisposing factor. Furthermore, molecular investigation of the related isolates for the verification of their resistome and the detection of mobile genetic elements, where the Fosfomycin ARGs may be located, is essential in order to identify and evaluate the mechanisms of their dissemination. Another important project is a circumstantial molecular investigation of the Fosfomycin ARGs, in order to verify the exact factors that could provoke the presence or absence of a resistant phenotype and determine the levels of resistance.

Data provided by research in these sectors could undoubtedly contribute to an in-depth comprehension of the phenomenon and indicate the requirements for possible surveillance and control measures.

## 5. Conclusions

There is a renewed interest in Fosfomycin in the last two decades, as high rates of non-susceptibility against the traditionally used factors appear. Its desirable properties and wide spectrum of bactericidal activity reinforce its potential as an alternative agent. However, Fosfomycin resistance emerges worldwide. The appearance of resistant isolates in companion animals, where the drug has not been widely used, is an even more disturbing fact, indicating the wide dissemination of MDR strains and the danger of circulation of these strains among pets, their owners, and the environment. The results of this review demonstrate the presence of relevant strains in dogs and cats and the fact that the cause of their spreading, could be the extended use of other, routinely used antibacterial agents, that promote the prevalence of MDR, epidemic strains among an animal population. As the relative data are yet limited, further and more extensive research is essential in order to identify the various aspects of the phenomenon and evaluate possible preventive or control measures.

## Figures and Tables

**Figure 1 vetsci-10-00337-f001:**
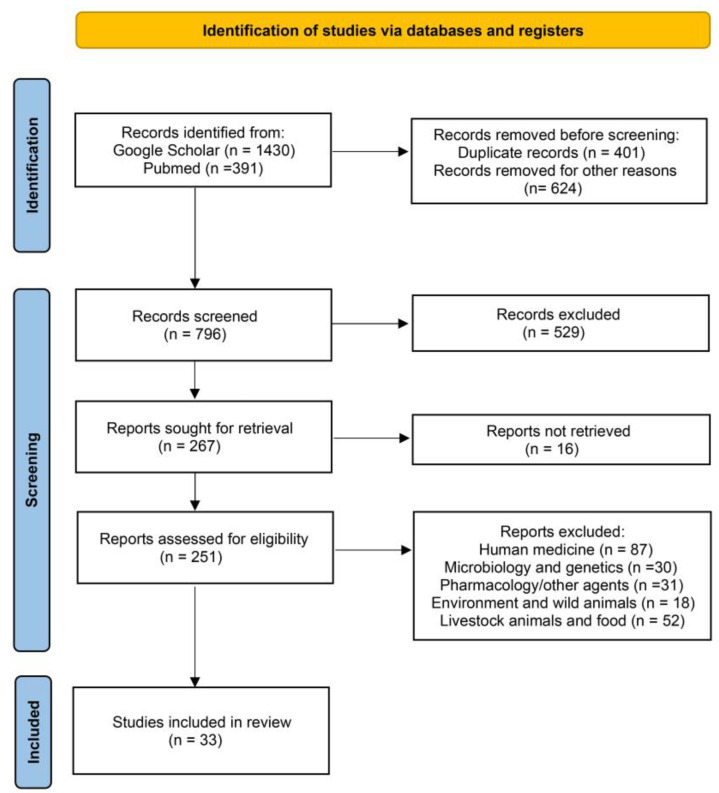
Identification of studies regarding Fosfomycin resistance in companion animals via databases using PRISMA guidelines [47].

**Figure 2 vetsci-10-00337-f002:**
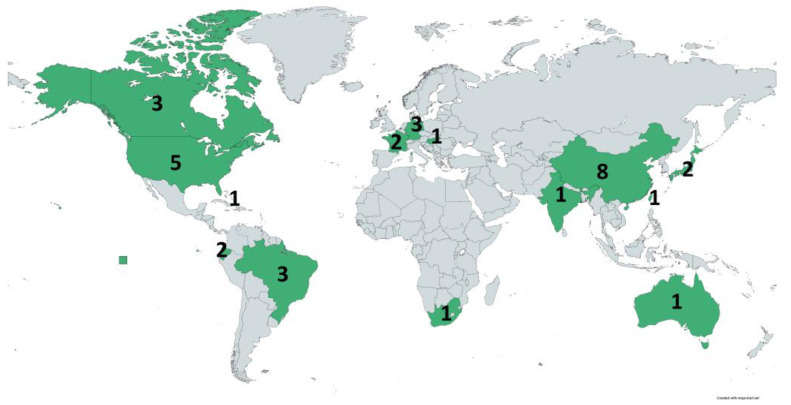
Distribution of the studies (countries in green) included in this review throughout the world. Numbers indicate the number of studies that investigated isolates from the corresponding country. (https://www.mapchart.net/world.html, accessed on 9 February 2023).

**Table 1 vetsci-10-00337-t001:** Main mechanisms of bacterial resistance against Fosfomycin.

Mechanism of Resistance	Description	Related Genes	References
Modification or overexpression of *MurA* gene	A modification of MurA gene, could alter amino acid sequence in Fosfomycin’s binding site, conferring resistance [28,29]. Furthermore, a resistant phenotype can also be achieved by increased synthesis of the enzyme through overexpression of the MurA gene.	*MurA*	[31,32]
Reduced permeability	Mutations affecting metabolic pathways of the membrane transporters responsible for the uptake of Fosfomycin (GlpT and UhpT) or their substrates, glycerol-3-P and glucose-6-P. Mutations affecting intracellular levels of cAMP-receptor protein (CRP), which affects the action of GlpT and UhpT.	Variable related genes	[31,33,34]
Fosfomycin modifying enzymes	*FosA* enzyme: a glutathione S-transferase that inactivates Fosfomycin by the addition of glutathione. Mn^+2^ and K^+^ are used as cofactors. Mostly found in Gram-negative bacteria such as Enterobacteriaceae and *Pseudomonas*. Several subtypes of *fosA* have been identified: *fosA3*, *fosA4*, *fosA5*, *fosA6*, *fosA8*, *fosA9*, *fosA10*, *fosA^EC^* and *fosA^SH^* ^1^.	*fosA*, *fosA3*, *fosA4*, *fosA5*, *fosA6*, *fosA8*, *fosA9*, *fosA10*, *fosA^EC^*, *fosA^SH^*	[35,36,37,38,39]
	*FosB* enzyme: differs from *fosA* in being a Mg^+2^ dependent enzyme and using l-cysteine or possibly bacillithiol, as the physiologic thiol donor. Additionally, an extracytoplasmic sigma factor *SigW*, seems to be essential for its expression. It is routinely detected in Gram-positive bacteria, (*Staphylococcus* spp., *Enterococcus* spp., *Bacillus subtilis*).	*fosB*	[36,40,41]
	*FosX* enzyme: a Mn^2+^-dependent epoxide hydrolase, which catalyzes the addition of a water molecule to C1 position of Fosfomycin’s oxirane ring and as a result breaks it and inactivates its antibacterial properties. It can be regularly found in specific species, such as *Listeria monocytogenes*, *Clostridium botulinum*, and *Brucella melitensis.*	*fosX*	[31,42]
	*FomA* and *fomB* enzymes: kinases that modify Fosfomycin by phosphorylation and thus detoxify it intracellularly. They are encountered in Fosfomycin-producing bacteria, such as *Streptomyces* spp.	*fomA*, *fomB*	[43]
	*FosC* enzyme: a kinase that converts Fosfomycin to Fosfomycin monophosphate, conferring intrinsic resistance in *Pseudomonas syringae*, through the expression of a chromosomally encoded gene.	*fosC*	[44]
Efflux pumps	*Tet38* (when overexpressed) and *AbaF* pumps in *Staphylococcus aureus and Acinetobacter baumannii*, respectively, contribute to Fosfomycin resistance, possibly by acting as efflux transporters of the agent.	*Tet38*, *abaF*	[45,46]

^1^ *FosA^EC^* and *fosA^SH^* were initially reported as *fosA2* and *fosA7* genes, chromosomal variants of *fosA* of *E*. *cloace* and *S*. *enterica* serovar Heidelberg, respectively. They were later named *fosA^EC^* and *fosA^SH^* in the newly proposed nomenclature [35].

**Table 2 vetsci-10-00337-t002:** Public information about the studies included in this review.

Country/Area	Bacterial Species	ARGs	Sample Origin	Fosfomycin Resistant/Total	Date/ Period	Reference
USA	*E. coli*	NS ^1^	Dogs, cats	3/275	2008–2010	[48]
France	MRSA	*fosB*	Dogs, cats	19/23 (19 *fosB*)	2006–2010	[49]
China	*E. coli*	*fosA3*	Dogs, cats	33/323 (29 *fosA3*)	2006–2010	[50]
China	*E. coli*	*fosA3*	Dog	1/1 (1 *fosA3*)	2008	[51]
China	*E. coli*	*fosA3*	Dogs, cats	12/766 (11 *fosA3)*	2008–2010	[52]
USA, Canada	MRSP ^2^	*fosB*	Dogs	7/31 (27 *fosB*)	NR ^3^	[53]
France	*P. aeruginosa*	NS	Dogs	22/46	2008–2011	[54]
Germany	*Ac. baumannii*	NR	Dogs, cats	25/25	2000–2013	[55]
China	Enterobacteriacae (*E. coli*, *Pr. mirabilis*, *E. fergusonii*, *C. freundii* *E. aerogenes*, *Kl. oxytoca*, *Kl. pneumoniae*	*fosA3*, *fosA*	Dogs, cats, pet owners	19/171 (16 *fosA3:* 8 *E. coli*, 4 *Pr. mirabilis*, 3 *E. fergusonii*, 1 *C. freundii* and 3 *fosA*: 1 *E. aerogenes*, 1 *Kl. oxytoca*, 1 *Kl. pneumoniae*)	2013	[56]
Japan	*P. aeruginosa*	NR	Dogs, cats	71/200	NR	[57]
Japan	Enterobacteriacae (381 *E. coli*, *50 Kl. pneumoniae*, *56 P. mirabilis*)	*fosA3*	Dog, cat	3/487 (3 *fosA3:* 2 *E. coli*, 1 *Kl. pneumoniae*)	2016	[58]
Canada	Enterobacteriacae (*Kl. pneumoniae*, *E. cloacae*)	*fosA*	Dogs	3/47 (3 *fosA:* 2 *Kl. pneumoniae*, 1 *E. cloacae*)	2015–2016	[59]
Germany	*Staphylococcus cohnii* subsp. *Cohnii*	NR	Dog	1/1	2015–2016	[60]
India	XDR ^4^ *E. coli*	NR	Dog	1/1	NR	[61]
Brazil	*P. aeruginosa*	*fosA*	Dog	1/1 (1 *fosA*)	2016	[62]
China	*E. coli*	NR	Dog	1/1	2013	[63]
Ecuador	*E. coli*	NR	Dogs	10/23	2017	[64]
Canada	*E. coli*, *St. pseudintermedius*	NF ^5^ (Absence of *fosA*, *fosA3*, *fosB*, *fosC2*)	Dogs	7/274, 4/113	2013–2016	[65]
Australia	Enterobacteriaceae, *P. aeruginosa*	*fosA*	Small Animal Hospital environment	65/656, 23/59	2016–2017	[66]
Brazil	*Kl. pneumoniae* (susceptible in the AST)	*fosA*	Dog	1/1 (1 *fosA*)	2018	[67]
Brazil	*Kl. pneumoniae* (susceptible in the AST)	*fosA*	Dog	1/1 (1 *fosA*)	2019	[68]
Taiwan	*Enterobacter cloacae*	*fosA3*, *fosA*	Dogs, cats	8/19 (8 *fosA*, *3* co-carried *fosA3*)	2010–2013	[69]
Ecuador	*E. coli*	*fosA3*	Dog	1/1 (1 *fosA3*)	2016	[70]
USA	*Salmonella* spp.	*fosA7* (*fosA^SH^*) ^6^	Dogs	2/27 (2 *fosA7)*	2013–2014	[71]
China	*Kl. pneumoniae*	*fosA*	Dogs, cats	105/105 (105 *fosA)*	2017–2019	[72]
Germany (China) ^7^	*Salmonella enterica* serovar Telelkebir	*fosA7* (*fosA^SH^*) ^6^	Dog	1/1 (1 *fosA7*)	2007	[73]
Hungary	*Staphylococcus aureus*	*fosB*, *murA* and *glpT* modification	Dog and owner	4/27 (2 *fosB*, *2 murA* and *glpT)*	NR	[74]
USA (China) ^7^	*Micrococcus luteus*	*murA*	Dog	1/1 (1 *murA*)	2019	[75]
China	*E. coli (mcr-1)*	*fosA3*	Dogs, cats	7/7 (7 *fosA3*)	2021	[76]
Caribbean	*Klebsiella pneumoniae*	*fosA*	Dogs, cats	2/2 (2 *fosA*)	2011–2018	[77]
South Africa	*E.coli* [MIC:(R), DD:(S)] ^8^	NR	Dog	1/1	NR	[78]
USA	*Staphylococcus aureus*	*fosB*	Dogs, cats	42/53 (42 *fosB*)	2017–2020	[79]
China	*Salmonella enterica* serovar Dublin	*fosX*	Dogs	NR	2018	[80]

^1^ NS: Not searched. ^2^ MRSP: Methicillin-Resistant *Staphylococcus pseudintermedius*. ^3^ NR: Not referred. ^4^ XDR: Extensively Drug-Resistant. ^5^ NF: Not found. ^6^ *FosA7* was later named *FosA*^SH^ in a newly proposed nomenclature [39]. ^7^ Germany and the USA were the countries where the bacteria were isolated, while China was the country of origin of these studies [73,75]. ^8^ Resistance when examined with the Minimum Inhibitory Concentration (MIC) method and susceptibility when examined with the Disc Diffusion (DD) method.

**Table 3 vetsci-10-00337-t003:** Microorganisms referred to in the selected studies.

Microorganisms	Number of References	Fosfomycin-Resistant Isolates *	Related Fosfomycin ARGs **
*E. coli*	13	86	*fosA3* (58)
*Kl. pneumoniae*	7	113	*fosA* (112), *fosA3* (1)
*P. aeruginosa*	4	117	*fosA* (24)
*St. aureus*	3	65	*fosB* (63), *murA* (2), *glpT* (2)
*Salmonella* spp.	3	3	*fosA^SH^* (3), *fosX*
*St. pseudintermedius*	2	11	*fosB* (7)
*Enterobacter cloacae*	2	9	*fosA* (9), *fosA3* (3)
*E. aerogenes*	1	1	*fosA* (1)
*Kl. oxytoca*	1	1	*fosA* (1)
*E. fergusonii*	1	3	*fosA3* (3)
*Pr. mirabilis*	1	4	*fosA3* (4)
*C. freundii*	1	1	*fosA3* (1)
*Micrococcus luteus*	1	1	*murA* (1)
*Ac* *. baumannii*	1	25	NR
*Staphylococcus cohnii*	1	1	NR

* study [66] was excluded from the analysis regarding Enterobacteriaceae, as they did not refer to specific numbers of isolates investigated per bacterial species. Study [80] does not refer to the specific number of *Salmonella* Dublin isolates carrying the *fosX* ARG. ** the number of isolates that were detected to carry each ARG is referred to in parentheses.

**Table 4 vetsci-10-00337-t004:** ARGs and resistance against other antibacterial agents, in the isolates included in this review.

Country /Area	Bacterial Species	Fosfomycin Related ARGs	Other ARGs with ≥50% Prevalence among Fos-Resistant Isolates ^1^	Agents with ≥50% Resistance Rates among Fos-Resistant Isolates ^1,2^	Reference
France	*S. aureus*	*fosB*	*mecA*, *blaZ*, *aadD*	ENR, ERY, FOX, KAN, LIN, PEN, SPI, TOB	[49]
China	*E. coli*	*fosA3*	*bla_CTX-M_*, *rmtB*	AMK, CHL, CTX, GEN, TET	[50]
China	*E. coli*	*fosA3*	*bla_CTX-M_*, *rmtB*	---	[51]
China	*E. coli*	*fosA3*	*bla_CTX-M_*	CHL, CIP, COT, GEN, NAL, TET	[52]
USA, Canada	*S. pseudi-ntermedius*	*fosB*	*mecA*	b-lactams ^3^	[53]
Germany	*A. baumannii*	NR	---	CXM, CFD, PIT, SAM	[55]
China	*E. coli*, *Pr. mirabilis*, *E. fergusonii*, *C. freundii*	*fosA3*	*bla_CTX-M_*	AMK, AMP, FAZ, CHL, CIP, GEN, FFC, KAN	[56]
Japan	*E. coli*, *Kl. pneumoniae*	*fosA3*	ESBL, *pAmpC*	---	[58]
Canada	*Kl. pneumoniae*, *E. cloacae*	*fosA*	*bla_CTX-M-15_*, *aac(3)-IIa*, *strA*, *strB*, *aac(6′)Ib-cr*, *bla_OXA-1_*, *bla_SHV-83_*, *bla_TEM-1-B_*, *qnrB1*, *sul2*, *drfA14*, *tetA*	---	[59]
Germany	*Staphylococcus cohnii*	NR	*mecA*	OXA	[60]
India	*E. coli*	NR	*bla_CTX-M_*, *bla_AmpC_**bla_TEM_*, *bla_NDM-1_*, *sul1*	AMK, AMC, AZT, CAZ, CFD, CFM, CHL, CIP, COL, CRO, CTX, CTC, CTR, ERT, FEP, FOX, GAT, GEN, IMP, MER, MOX, NOR, NIT, OFL, PMB, SXT, TET, TOB, VAN	[61]
Brazil	*P. aeruginosa*	*fosA*	*bla_VIM-2_*, *bla_PAO_*, *bla_OXA-4_*, *bla_OXA-50_*, *aadA2*, *aac(3)-Id*, *aph(3)-IIb*, *catB7*, *cmlA1*, *sul1*, *dfrB5*, *tetG*	AMK, AMC, CAZ, CIP, CHL, CRO, CTX, FEP, FOX, GEN, IMP, MER, NAL, PIT, STX, TET, TIC	[62]
China	*E. coli*	NR	*bla_NDM-1_*, *drfA17*, *sul1*, *aadA5*	CAZ, CTX, CIP, ERT, GEN, IMP, MER, PIP, TET	[63]
Ecuador	*E. coli*	NR	*bla_CTX-M_*	AZT, CAZ, CHL, CIP, CTX, DOX, FEP, LEV, NAL, NOR, TET	[64]
Canada	*S. pseudinter-* *medius*	NF	*mecA*	PEN, OXA, AMP, CLI	[65]
Brazil	*Kl. pneumoniae*	*fosA*	*bla_CTX-M-15_*, *bla_SHV_*, *bla_OXA-1_*, *aph(3”)-Ib*, *aph(6)-Id*, *aac (3)-IIa*, *tetA*, *aac(6′)*-*Ib-cr*, *qnrB1*, *oqxA and oqxB*, *dfrA*	AMC, CAZ, CIP, CTX, ENR FEP, FUR, GEN, LEV, NOR, OFL, TET	[67]
Brazil	*Kl. pneumoniae*	*fosA*	*bla_CTX-M-15_*, *bla_SHV_*, *bla_OXA-1_*, *1*, *aadA2*, *aph(3′)-Ia]*, *mphA*, *catB3*, *aac(6′)Ib-cr*, *oqxA*, *oqxB]*, *sul1*, *tetA*, *dfrA12*, *GyrA*, *ParC*	AMC, AZT, CIP, CRO, ENR, FEP, FUR, LEV, NAL, SXT, TET	[68]
Taiwan	*Enterobacter cloacae*	*fosA3*, *fosA*	*bla _TEM_*	AMP, SUD	[69]
Ecuador	*E. coli*	*fosA3*	*mcr-1.1*, *bla_CTX-M-3_*, *bla_TEM-206_*, *bla_TEM-1B_*, *tetA*, *GyrA*, *ParC*	CIP, COL, CRO, FEP, SAM	[70]
China	*Kl. pneumoniae*	*fosA*	*bla_SHV_*, *oqxA*, *oqxB*, *sul*	AMC, DOX, FFC, SXT	[72]
Germany (China)	*Salmonella enterica*	*fosA7 (fosA^SH^)*	*aac(6′)-Iaa_1*	---	[73]
Hungary	*Staphylococcus aureus*	*fosB*, *murA*, *glpT*	*blaZ*	PEN	[74]
China	*E. coli*	*fosA3*	*aac(3)-IVa*, *aph(3′)-IIa*, *aph(3′)-Ia*, *and aph(4)-Ia*, *bla_CTX-M-65_*, *bla_TEM-1B_*, *floR*, *drfA14*, *mcr-1*, *sul2*, *qnrS1*, *mdfA*	COL, CTX, FFC	[76]
South Africa	*E.coli*	NR	---	AMC, AMP, CEP, ENR, FUR, NEO, PEN	[78]

^1^ In case of one relevant strain in an article, ARGs and phenotypic resistances are referred to this isolate. ^2^ Antibacterial agents: AMC: Amoxicillin-clavulanate, AMK: Amikacin, AMP: Ampicillin, AZT: Aztreonam, CAZ: Ceftazidime, CEP: Cephalothin, CFM: Cefixime, CHL: Chloramphenicol, CFD: Cefpodoxime, CIP: Ciprofloxacin, COL: Colistin, CPR: Cefoperazone, CRO: Ceftriaxone, CTC: Cefotaxime-clavulanate, CTX: Cefotaxime, CXM: Cefuroxime, DOX: Doxycycline, ENR, Enrofloxacin, ERT: Ertapenem, ERY: Erythromycin, FEP: Cefepime, FFC: Florfenicol, FOX: Cefoxitin, FUR: Ceftiofur, GAT: Gatifloxacin, GEN: Gentamicin, IMP: Imipenem, KAN: Kanamycin, LEV: Levofloxacin, LIN: Lincomycin, MOX: Moxalactam, NAL: Nalidixic acid, NEO: Neomycin, NOR: Norfloxacin, NIT: Nitrofurantoin, OFL: Ofloxacin, OXA: Oxacillin, PEN: Penicillin PMB: Polymyxin B, PIP: Piperacillin, PIT: Piperacillin-tazobactam, SAM: Ampicillin-sulbactam, SPI: Spiramycin, SXT: Sulfamethoxazole-Trimethoprim, SUD: Sulfadiazine, TET: Tetracycline, TIC: Ticarcillin, TOB: Tobramycin, VAN: Vancomycin. ^3^ Except Ceftaroline.

## Data Availability

No new data were created or analyzed in this study. Data sharing is not applicable to this article.

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
