# Peer review of "Fosfomycin Resistance in Bacteria Isolated from Companion Animals (Dogs and Cats)"

_vetsci, 2023, doi:10.3390/vetsci10050337_

Round 1
Reviewer 1 Report
All corrections should be done before start of publication process
There is no conclusion
There is no highlights----should be
The abstract needs more improvement and should contained backgrounds(including aims) , Methods , Results and conclusion
In this review , the authors used a huge a mounts of abbreviations:-
a-Should create a table for all of this
b-Should firstly detailed then abbreviate
There is no recommendations
There is ethical approval state
There is no acknowledgement
LN/9-21----the simple summary should move to the end of the review with conclusion or as a conclusion after treatment
The authors did not mention anything about the Fosfomycin-urinary tract infection -humans usage----either in abstract /introduction
LN/23----canine and feline----particularly ---why ????
LN/23---PRISMA---give detail then abbreviate
LN/36---add PRISMA guidelines and ARGs to the keywords
LN/42----add reference
From LN/40--------------till LN/56---the authors again running behind against what not for what ---even he will mention later
LN/49-50----bacterial cell wall biogenesis----what is the pathogenesis for this ?????
LN/51---GLpT,UhpT---for ordinary readers ----give more details
LN/59---add pus instead of abscess fluid
LN/59---as anti-inflammatory---explain how ???
LN/59-62---not clear ---readjust it
Introduction is very long ----be more concise
Clear the aims of the study at the end of introduction(I)
LN/197---add public instead of generic
There is no statistical analysis at materials and methods
All data under tables should be summarize
From LN/502----510---not discussion
Discussion should be more concise and based upon debating the obtained results with those of the previous investigators results
In a simple manner----can tell how can solve this big problem(antibiotic-strains-resistance) ???
As volume , issue , number and pages ---all are available----so no need for the link(s)---APPLY FOR ALL
Some cited references are very old ----update all
The most used references contained more than 6 authors ----why ??? should be 6 at the maximum plus etal with the last ones ---APPLY FOR ALL
LN/663---why all capitals ??
LN/701-702----write as 13 :1273-1279
LN/786--delete pp --apply for all
What a huge number of references used (114) ????

Author Response
- There is no conclusion
Answer: Even though the Conclusions section is considered optional for Veterinary Sciences MDPI, we agree with the reviewer. We excluded the last paragraph from the discussion section, modified it and added a «Conclusions» section.
- There is no highlights----should be
Answer: Even though a highlights sections is not included in the “instructions for Authors” for Veterinary Sciences MDPI, we have prepared highlights:
- coli is the predominant bacterium isolated from pets, with acquired Fosfomycin-resistance.
- FosA and fosA3 are the ARGs mostly detected in gram-negative bacteria and fosB in gram-positive.
- The grand majority of Fosfomycin-resistant isolates from pets are MDR.
- Both Fosfomycin-resistant Staphylococci and Enterobacteriaceae, demonstrate usually a resistant phenotype against β-lactams, mediated by mecA and ESBL genes respectively.
- Fosfomycin resistance is possibly disseminated among pets through the usage of other classes of antibiotics, such as β-lactams and aminoglycosides.
We leave this at the Editor’s decision if these highlights can be added as a separate section in the manuscript.
- The abstract needs more improvement and should contained backgrounds (including aims) , Methods , Results and conclusion
Answer: We agree with the reviewer, the Abstract was modified accordingly in a single paragraph, with the style of a structured abstract, but without headings, as referred in the “Instructions for Authors”.
- In this review , the authors used a huge amounts of abbreviations:-
a-Should create a table for all of this
b-Should firstly detailed then abbreviate
Answer: We agree with the reviewer. The appropriate modifications were carried out in the article (first detailed terms – then abbreviated). If the reviewer thinks necessary, we can add a table in supplementary file with all abbreviations as the following:
|
Abbreviation |
Detailed Meaning |
|
ARG |
Antibiotic Resistance Gene |
|
MDR |
MultiDrug Resistant |
|
PRISMA |
Preferred Reporting Items for Systematic Reviews |
|
GlpT |
Glycerol-3-phosphate (G-3-P) Transporter |
|
UhpT |
Hexose phosphate uptake transporter /Glucose-6-phosphate transporter |
|
cAMP |
Cyclic Adenosine Monophosphate |
|
CRP |
cAMP – Receptor Protein |
|
ESBL |
Extended Spectrum B-Lactamase |
|
VRE |
Vancomycin Resistant Enterococci |
|
CR |
Carbapenem Resistant |
|
MRSA |
Methicillin Resistant Staphylococcus aureus |
|
MRSP |
Methicillin Resistant Staphylococcus pseudintermedius |
|
XDR |
Extensively Drug Resistant |
|
MIC |
Minimum Inhibitory Concentration |
|
DD |
Disc Diffusion |
|
MRR |
Multiresistance Region |
|
NDR |
No Drug Resistance |
|
SDR |
Single Drug Resistance |
|
NCBI |
National Center for Biotechnology Information |
|
UTI |
Urinary Tract Infection |
|
ICU |
Intensive Care Unit |
|
PDR |
Pun-drug Resistant |
|
CPKP |
Carbapenemase-Producing Klebsiella pneumoniae |
- There is no recommendations
Answer: We agree with the reviewer, recommendations were added in a specific paragraph at the last paragraph of the Discussion.
- There is ethical approval state
Answer: According to instructions for authors, ethical approval statement is included in the “Institutional Review Board Statement:” if applicable. However, the study did not involve humans or animals, so the “Not applicable” statement is reported.
- There is no acknowledgement
Answer: We agree with the reviewer, acknowledgement section was added, even though “Not applicable” is reported.
- LN/9-21----the simple summary should move to the end of the review with conclusion or as a conclusion after treatment
Answer: The Simple summary section was included in the relevant MDPI template before the Abstract section. The study team used the MDPI Microsoft Word template as downloaded by https://www.mdpi.com/journal/vetsci/instructions#front .If the Editor agrees, we could move the Simple summary at the end of the review.
- The authors did not mention anything about the Fosfomycin-urinary tract infection -humans usage----either in abstract /introduction.
Answer: We thank the reviewer for the comment. In the introduction section, it is now referred that: «Oral Fosfomycin is primarily used in human medicine, in cases of uncomplicated urinary tract infections (UTIs) and prostatitis caused by multidrug-resistant (MDR) Gram-negative bacteria.» In order to furtherly emphasize this, a specific reference was added in the beginning of the Abstract, please see first sentence.
- LN/42----add reference
Answer: We agree with the reviewer. This reference was added (Reference N. 1).
- LN/23----canine and feline----particularly ---why ????
Answer: We thank the reviewer for the comment. These are the only companion animals with relevant data in the current literature, sufficient to support the purpose of a review. Their household accommodation and the closer contact with their owners, in contrast with other species, emphasizes the public health issue which is pointed out in this review. In order to clarify this, a change to the title is proposed, adding “dogs and cats”.
- LN/36---add PRISMA guidelines and ARGs to the keywords
Answer: We agree with the reviewer. PRISMA guidelines and ARGs were added to the keywords.
- From LN/40--------------till LN/56---the authors again running behind against what not for what ---even he will mention later
Answer: We agree with the reviewer. The whole part of the Introduction was modified in order to re-organize the text and make it more concise and comprehensive, according to the reviewers’ comments.
- LN/49-50----bacterial cell wall biogenesis----what is the pathogenesis for this ?????
Answer: We agree with the reviewer. The text was modified in order to more clearly note the mode of the bactericidal effect of the agent.
- LN/23---PRISMA---give detail then abbreviate
Answer: We agree with the reviewer, the respective modification was carried out in the Abstract.
- LN/59---add pus instead of abscess fluid
Answer: We agree with the reviewer, the respective modification was carried out.
- LN/59-62---not clear ---readjust it
Answer: We agree with the reviewer, the sentence was modified in order to become clear.
- Introduction is very long ----be more concise
Answer: We agree with the reviewer, the size of the Introduction was reduced by modifying the text in several points and including the resistance mechanisms in a respective table (please see Table 1) and some related references were excluded.
- Clear the aims of the study at the end of introduction(I)
Answer: We agree with the reviewer, the aims of the study were clearly noted, as the text was modified accordingly at the end of the Introduction.
- LN/197---add public instead of generic
Answer: We agree with the reviewer, the respective modification was carried out.
- There is no statistical analysis at materials and methods
Answer: We thank the reviewer for the comment, we added a relevant comment at the end of the Materials and Methods section.
- All data under tables should be summarized
Answer: We agree with the reviewer. A column was added in Table 2 and in Table 3 with numbers of samples Fosfomycin resistant / total and the gene positive strains in parenthesis as well as total Fosfomycin resistant isolates and resistance genes carried per bacterial species.
- From LN/502----510---not discussion
Answer: We agree with the reviewer. This part was excluded from the discussion, which was modified according to the comments 23 and 24.
- Discussion should be more concise and based upon debating the obtained results with those of the previous investigators results. In a simple manner----can tell how can solve this big problem(antibiotic-strains-resistance) ???
Answer: We agree with the reviewer. We modified the whole Discussion part, in order to emphasize the connection between the findings and the current literature, to analyse the danger and the possible causes of the phenomenon and finally to recommend subjects for future research which could provide sufficient data for the proper preventive or control measures.
- As volume , issue , number and pages ---all are available----so no need for the link(s)---APPLY FOR ALL
- Some cited references are very old ----update all
- The most used references contained more than 6 authors ----why ??? should be 6 at the maximum plus etal with the last ones ---APPLY FOR ALL
- LN/663---why all capitals ??
- LN/701-702----write as 13 :1273-1279
- LN/786--delete pp --apply for all
- What a huge number of references used (114)
Answer: We agree with the reviewer regarding all these comments (25-31). The aforementioned references were corrected accordingly, the old cited references were updated when it was possible and their total number was reduced.

Reviewer 2 Report
Marios Lysitsas and colleagues (vetsci-2348237) made a significant effort for combined analysis of Fosfomycin resistance in bacteria isolated from companion animals over the world. However, there should be certain points that might improve the manuscript.
The introduction part is complicated, I would suggest using a figure and table to summarize the content.
Meta-analysis is usually linked with a systematic review. Is there a quantitative analysis that can be used in this study? considering the data is available.
The quality of figure 1 should be improved, there is a grey line on the left.
Lines 211-216. This information could be marked in figure 2.
Lines 521-528. These points should be subjected to text. Same issue for Lines 545-565.
One relevant literature dealing with Salmonella resistance is missing, here is the DOI number (10.3390/antibiotics11050625).
The structure of the manuscript is confusing, how the review has a short discussion part? The majority part is about the author's idea, so it should be indicated as the summary or conclusion part.
Last but not least, I would like to see more knowledge or information regarding Fosfomycin usage in pets, I also disagree with the point that "Since Fosfomycin is an agent not routinely used by veterinarians, ". Here in some countries, they are still routinely used against diarrhea. Did you have enough information regarding worldwide usage for this particular drug of interest in this study?
Additionally, other mechanisms, such as the pump, may also play a role in developing resistance, which is missing in this study.
Nevertheless, a re-organization of the structure should be made, and more knowledge regarding usage and mechanism should be supplemented.
Some minor points after re-organizing the manuscript.
Author Response
Reviewer 2
- The introduction part is complicated, I would suggest using a figure and table to summarize the content.
Answer: We agree with the reviewer. The text was modified in order to be more concise and well-organised, and a table was used to replace the part with the resistance mechanisms.
- Meta-analysis is usually linked with a systematic review. Is there a quantitative analysis that can be used in this study? considering the data is available.
Answer: We thank the reviewer for the comments, we have added quantitative results: Numbers of isolates reported to be Fosfomycin resistant/total number of isolates investigated and numbers of ARGs detected, adding also comments to the discussion on these results.
- The quality of figure 1 should be improved, there is a grey line on the left.
Answer: We agree with the reviewer, the figure was modified as a new article was included (comment 6) and the gray line was deleted.
- Lines 211-216. This information could be marked in figure 2.
Answer: We agree with the reviewer we added the relevant data of number of studies that investigated isolates originating from the corresponding country.
- Lines 521-528. These points should be subjected to text. Same issue for Lines 545-565.
Answer: We agree with the reviewer, the text was modified accordingly.
- One relevant literature dealing with Salmonella resistance is missing, here is the DOI number (10.3390/antibiotics11050625).
Answer: We agree with the reviewer. The relevant article was included (it was missed by mistake during the assessment stage from the review).
- The structure of the manuscript is confusing, how the review has a short discussion part? The majority part is about the author's idea, so it should be indicated as the summary or conclusion part.
Answer: We agree with the reviewer. The discussion part was modified, we tried to present a more clear correlation of the findings with those of other studies and reduce the part about the author’s idea. Additionally, a reorganization of this part was attempted.
- Last but not least, I would like to see more knowledge or information regarding Fosfomycin usage in pets, I also disagree with the point that "Since Fosfomycin is an agent not routinely used by veterinarians, ". Here in some countries, they are still routinely used against diarrhea. Did you have enough information regarding worldwide usage for this particular drug of interest in this study?
Answer: We agree with the reviewer. The usage of Fosfomycin is rather limited and disapproved in many countries, but not all and therefore we modified the text in the respective parts, making that more clear. For example “Regarding companion animals, even if the use of Fosfomycin is relatively limited in many countries…” etc. Moreover we added some more information in the part of the text referring to the veterinary usage of the antibiotic (part 1.3.) and three recent associated references (two of them are documents from WHO and European Medicines Agency in order to support the relatively limited use in EU).
- Additionally, other mechanisms, such as the pump, may also play a role in developing resistance, which is missing in this study.
Answer: We agree with the reviewer. Efflux pumps, even though not the main resistance mechanisms against Fosfomycin, may contribute to resistance. Therefore, they were included to the relevant table in the Introduction part and the respective references we detected (42 and 43) were added too.
- Nevertheless, a re-organization of the structure should be made, and more knowledge regarding usage and mechanism should be supplemented.
Answer: A re-organization of the text in the Introduction part was accomplished and we tried to be more concise and comprehensive regarding the mechanism of action, the properties of the agent and its usage.
Round 2
Reviewer 2 Report
none
none